# Constraining Earth's core composition from inner core nucleation

Alfred J. Wilson [1] ✉, Christopher J. Davies [1], Andrew M. Walker [2] & Dario Alfè [3,4,5]

The composition of Earth's core is a fundamental property of the Earth's deep interior, defining its present structure and long term thermal and magnetic evolution. However, the composition of the core is not well understood, with several combinations of light elements being able to satisfy the traditional constraints from cosmochemistry, core formation and seismology. The classic view of inner core formation does not include the necessity for liquids to be supercooled to below their melting point before freezing. Attempts to calculate the magnitude of this supercooling have found that several binary core compositions are incompatible with inner core nucleation. Here we show, through molecular dynamics simulations, that nucleation from an $Fe_{1-x}C_{x=0.1-0.15}$ composition is compatible with a range of geophysical constraints. Whilst not a complete description of core chemistry, our results demonstrate that inner core nucleation places a strong constraint on the composition of Earth's core that may allow discrimination between previously identified potential compositions.

The composition of Earth's iron-rich core plays a crucial role in determining the structure, dynamics and evolution of Earth's deep interior. The melting temperature, $T_m$, of the core, set by the constituent iron alloy, defines the temperature, $T$, at the inner core boundary (ICB), which provides a unique constraint on the present-day temperature at the core mantle boundary[1] (CMB). Transport properties also vary with composition, including thermal conductivity, $k$, which plays a vital role in determining thermal stratification and convective vigour in the liquid core and therefore the long term thermal evolution of the core[1–3] and its potential to generate the global magnetic field, which has been preserved in the rock record for at least the past 3.5 Gyrs[4–6]. Light elements determine the strength of compositional buoyancy produced by inner core growth (the dominant power source for the geodynamo today[1,7,8]) through their partitioning between solid and liquid during inner core freezing[9], while chemical exchange at the CMB may produce stable regions at the top of the core[10–12] that are detectable by seismology[13–15]. However, despite recent progress[16], the composition of the Earth's core remains poorly known.

Three main approaches have been used to constrain the composition of Earth's core: cosmochemistry, core formation and seismology. Core composition can be inferred via cosmochemistry by comparing the composition of primitive CI meteorites, those which most closely resemble the solar photosphere, with the silicate Earth. CI meteorites are rich in Fe, Ni, Mg, Ca, Al, Si, S, C and O[17], where the light elements Si, S, C and O are appealing candidates to explain the low density of the core compared to pure Fe[18]. If the Earth is assumed to be assembled primarily from CI meteorites, deviations of the bulk silicate Earth from their composition can be ascribed to losses to space or the core. This approach favours an Fe-Ni (~85 wt% and ~5 wt %, respectively) core[19] where Si is the major light element (up to 9.6 wt %[20]) and C, S, and P cumulatively make up 2.5 wt% of the core[19]. Core formation models estimate core composition by assuming chemical equilibrium between metal and silicate during Earth's accretion and differentiation. The equilibrium concentration of light elements depends on partition coefficients, determined by experiments and calculations, which vary with pressure, temperature and composition.

[1]School of Earth and Environment, University of Leeds, Leeds, UK. [2]Department of Earth Sciences, University of Oxford, Oxford, UK. [3]Department of Earth Sciences, University College London, London, UK. [4]London Centre for Nanotechnology, University College London, London, WC1H 0AH, UK. [5]Dipartimento di Fisica "Ettore Pancini", Universita' di Napoli "Federico II"m, Napoli, Italy. ✉ e-mail: a.j.wilson1@leeds.ac.uk

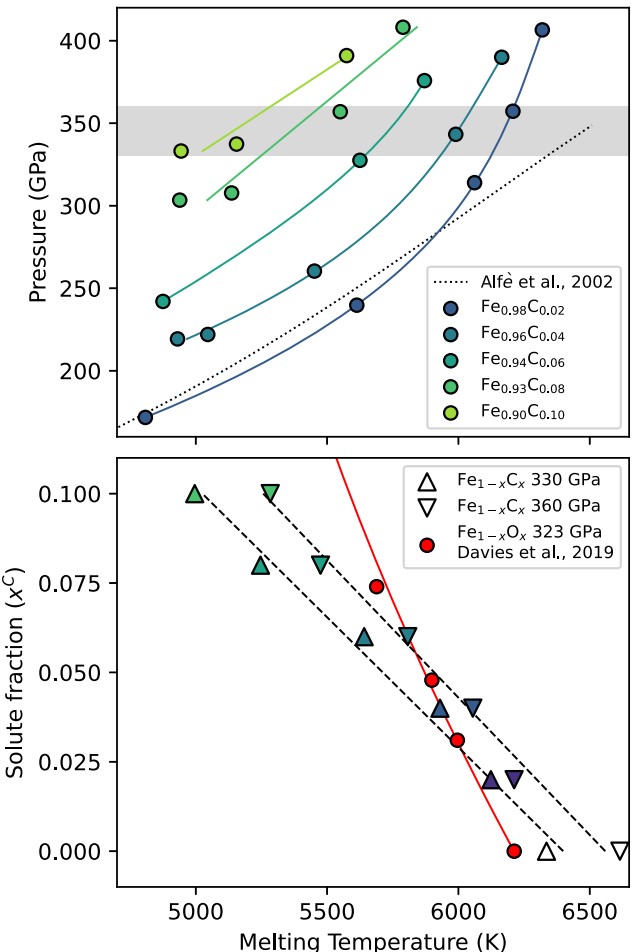

**Fig. 1 | Melting temperatures of Fe–C alloys at core conditions.** Upper: Melting temperatures (points) calculated using two-phase coexistence simulations of $Fe_{1-x}C_x$ systems (where $x$ is molar fraction). The dotted line shows the melting curve of pure Fe from Alfè et al.[36] for reference. Solid lines are fits to data (2nd degree polynomial for $x^C = 0.02, 0.04, 0.06$ and linear for $x^C = 0.08, 0.10$). The grey shaded region shows the $P$ range of the Earth's inner core. Lower: Interpolation of points in the upper panel gives $T_m(330\,GPa, x^C)$ and $T_m(360\,GPa, x^C)$, shown as up and down pointing triangles, respectively. These conditions represent the present-day ICB and the centre of Earth, respectively. The $Fe_{1-x}O_x$ result of Davies et al.[30] at 323 GPa is shown for comparison (red points and line).

The accretionary history of the Earth is therefore a key component in the determination of core composition. Several models[21–23] favour high Si and O (7.1–9.9 wt% and 1.3–5.3 wt% respectively) in the core. However, the array of accretionary histories that are possible results in a wide range of plausible compositions[24]. Comparison of the elastic properties of the core from seismology with results from mineral physics can be used to identify core compositions that are compatible with observations. The depth varying wave speeds of the outer and inner core and the density contrast across the ICB[25] require ternary systems, although several combinations and concentrations of C, O, Si and S are viable[16]. For example, Badro et al.[26] find that whilst an Fe-Ni core with 3.7 wt% O and 1.9 wt% Si best satisfies the available constraints, other combinations of O with C, Si or S can also match the radial P-wave structure of the core. Ultimately, no single composition is uniquely able to explain the origins, formation and elastic properties of the Earth's core and the range of plausible compositions have markedly different implications for the thermal state of the core, both past and present. Given these uncertainties, it is important to seek additional constraints on core composition that are independent

from but complementary to existing approaches. Here, we propose that a constraint on core composition can be derived by analysing the thermodynamic conditions under which the solid inner core first formed.

The Earth's inner core is classically understood to have formed when the $T$ of the core at the centre of the Earth cooled to the $T_m$ at the same location. The composition of the core from this time onwards can be used to estimate the thermal profile of the deep Earth by assuming that the adiabatic temperature, $T_a$, of the core must equal the $T_m$ at the ICB[27]. However, this picture is physically incomplete because all liquids must be supercooled by an amount $\delta T$ below $T_m$ ($\delta T = T_m - T$), often significantly, before freezing can begin[28]. This requirement arises because whilst the solid phase is thermodynamically favoured for $T < T_m$, establishing an interface between solid and liquid is always unfavourable, and for the first solids, the energy change of introducing an interface always wins out over the phase change. Previous studies[29–34] have estimated the $\delta T$ required to nucleate the solid inner core for several compositions that are compatible with traditional constraints from core formation and seismology, but found that $\delta T$ is incompatible with geophysical constraints (see Wilson et al.[35] for a review). When considering the effect of light elements, compositions including O and C were found to require $\delta T$ closer to geophysically compatible values when compared to pure Fe, Fe–S and Fe–Si[33]. Because not all potential compositions of the core can explain the presence of the inner core, inner core nucleation may provide an additional and strong constraint on the composition of the core.

In this study, we extend our previous results on the Fe–C system, for which the required $\delta T$ for inner core nucleation is the closest to geophysical constraints of all systems tested previously[33]. We use an improved embedded atom model (EAM), validated at a higher C concentration of 10 mol%, to explore solid nucleation in molecular dynamic simulations of supercooled $Fe_{(1-x)}C_x$ liquids and establish the required conditions for the inner core to freeze from these liquids. Our calculations address homogeneous nucleation, where solids form spontaneously, away from any pre-existing solid surfaces. We return to consider heterogeneous nucleation, which arises in the presence of solid surfaces, in the discussion.

## Results

We use classical molecular dynamics (CMD) to observe and characterise the nucleation of atomic-scale solids in supercooled $Fe_{(1-x)}C_x$ liquids at core pressures. Classical nucleation theory (CNT)[28] states that the nucleation rate, $I$, of a system is inverse to the waiting time to observe nucleation, $\tau_w$, and increases with supercooling below the melting temperature.

### Melting temperatures

To characterise nucleation in molecular dynamic simulations for a specific $\delta T$, we require knowledge of the melting temperature in order to define an appropriate simulation $T$. Melting temperatures, $T_m$, are calculated using two-phase coexistence simulations for compositions between $Fe_{0.98}C_{0.02}$ and $Fe_{0.9}C_{0.1}$, shown in Fig. 1. Simulations are conducted at a range of $T$, volume, $v$, conditions, spanning the pressure, $P$, range of the inner core (330–360 GPa). At least 50 unique calculations are performed at each condition to ensure a sufficient number of configurations are sampled (see "Methods" for details). At low $P$ and carbon concentration, $x^C$, $T_m$ is comparable to the pure Fe case of Alfè et al.[36], the EAM of which is used for the Fe component of the model used in this study. At high $P$ and low $x^C$, $T_m$ depression is smaller than $Fe_{1-x}O_x$ of the same $x^C$ (albeit at slightly lower $P$). $T_m$ is depressed by a greater amount at high $x^C$, -1300 K at $x^C = 0.1$ and 330 GPa. Interpolation of results provides melting temperatures at 360 GPa, shown in Table 1.

**Table 1 | Parameters describing the nucleation rates and waiting times for spontaneous freezing of supercooled $Fe_{1-x}C_x$ liquids**

| | Wilson et al. (2023) | | This study | |
|---|---|---|---|---|
| | $Fe_{0.99}C_{0.01}$ | $Fe_{0.97}C_{0.03}$ | $Fe_{0.95}C_{0.05}$ | $Fe_{0.90}C_{0.10}$ |
| $N$ (m$^{-3}$) | $6.8 \times 10^{-35}$ | | $2.3 \times 10^{-34}$ | $2.8 \times 10^{-34}$ |
| $S$ (s$^{-1}$) | $5 \times 10^{13}$ | | $1.2 \times 10^{12}$ | $1.6 \times 10^{12}$ |
| $\tau_0$ (s m$^{-3}$) | $2.93 \times 10^{-23}$ | $4.63 \times 10^{-23}$ | $6.48 \times 10^{-23}$ | $1.51 \times 10^{-22}$ |
| $h_f$ (J m$^{-3}$) | $0.57 \times 10^{10}$ | $1.30 \times 10^{10}$ | $1.35 \times 10^{10} \pm 2 \times 10^9$ | $1.55 \times 10^{10} \pm 2.5 \times 10^9$ |
| $h_c$ | $1 \times 10^{-3}$ | $1 \times 10^{-6}$ | $1 \times 10^{-6} \pm 5 \times 10^{-7}$ | $1 \times 10^{-6} \pm 5 \times 10^{-7}$ |
| $\gamma$ (J m$^{-2}$) | 1.005 | 1.005 | $1.005 \pm 0.01$ | $1.005 \pm 0.004$ |
| $T_m$ (K) | 6444 | 6348 | 5866 | 5338 |

See "Methods" for details.

## Nucleation of iron−carbon alloys

We use CMD simulations of supercooled iron alloys to study the nucleation of solids. These simulations are independent of CNT; however, CNT provides an intuitive physical picture with which to interpret the simulation results and has been shown to accurately describe homogeneous nucleation in binary alloys at core conditions[33]. From our simulations, we obtain the nucleation rate, $I(r)$, directly for sub-critical nuclei and using CNT, we are then able to fit for the critical nucleus size, $r_c$, which has a 50% chance of spontaneously freezing a system, informing the average waiting time, $\tau_w$, to observe the freezing of a system (see "Methods"). This approach means that systems with low supercooling, and therefore small $I(r)$ and large $\tau_w$, can be studied directly, avoiding the large extrapolation necessary in prior approaches[30]. Furthermore, recording $I(r)$ in simulations makes no assumption of the nucleating phase or system behaviour, meaning that the properties of the system can be inferred provided that CNT accurately describes $I(r)$.

Critical radii, $r_c$, are estimated from $I(r)$ recorded from CMD simulations (see "Methods" and Wilson et al.[31] for details) at selected $T$ and $x^C$ and are shown in Fig. 2 with comparison to prior results for $x^C = 0.01$ and $x^C = 0.03$ from Wilson et al.[33]. Nuclei are approximately spherical for $r > 2$ Å and have a crystal structure best described as defect-rich hexagonal close packed, as was found in our previous study[33]. The average waiting time, $\tau_w$, is calculated as $\tau_w = \tau_0 \exp\left(\frac{\Delta G(r_c)}{k_B T}\right)$, where $\Delta G = \frac{4}{3}\pi r^3 g^{sl} + 4\pi r^2 \gamma$, $g^{sl}$ is the free-energy difference between solid and liquid phases, $\gamma$ is the interfacial energy at the boundary between solid and liquid, $k_B$ is the Boltzmann constant and $\tau_0$ is a kinetic prefactor ($\tau_0 = \frac{z}{NS}$) linked to the Zeldovich factor, the probability of freezing or growing a nuclei $z$, the number density of nucleation sites, $N$, and the growth rate of nuclei, $S$, (all of which are calculated from simulations). The remaining quantities required to calculate $\tau_w$ ($h_f$, $h_c$ and $\gamma$) are fit from $r_c(T) = \frac{-2\gamma}{h_f \frac{\delta T}{T_m}(1 - h_c \delta T)}$, at each $x^C$ (Fig. 2, see "Methods" for details).

The interatomic potential developed here reproduces the $r_c$ result of Wilson et al.[33] at $x^C = 0.01$ and 5000 K within 4% ($r_c = 9.16 \pm 1.86$ Å compared to $r_c = 9.52 \pm 2.31$ Å[33]). At all tested values of $\delta T$, increasing $x^C$ reduces $r_c$, although at large $\delta T$, the $r_c$ for all compositions are within the uncertainty of one another. Simulations with $x^C > 0.1$ show apparent phase separation with distinct high ($x^C > 0.3$) and low C regions, either due to immiscibility or limitations of the EAM potential. These high $x^C$ regions are beyond the conditions sampled by our ab initio calculations and so cannot be validated, we therefore omit simulations with $x^C > 0.1$ from our results.

Values of $\tau_w$ are shown in Fig. 3. Results for $x^C = 0.01$ and 0.03 are from Wilson et al.[33], while results for $x^C = 0.05$ and 0.10 are calculated from the quantities shown in Table 1. $N$ and $S$ are calculated as averages from nuclei distributions and allow calculation of $\tau_0$ (see "Methods"). $\tau_0$ is not found to vary with supercooling by more than one order of magnitude and so is taken as isochemical averages. Estimates of $\tau_w$ are compared to the value $3.1 \times 10^{34}$ s m$^{-3}$ (black dashed line, Fig. 3), which

implies that a region with half of the present-day inner core radius was supercooled for 1 Gyrs prior to nucleation[33]. The remaining half of the inner core radius would then have grown at a rate proportional to the cooling rate of the core after nucleation. A more extreme case, which maximises waiting time by requiring that the entire volume of the inner core was supercooled prior to nucleation, would be where the inner core has not grown since nucleation and the entire inner core volume was supercooled ($2.4 \times 10^{35}$ s m$^3$). The corresponding nucleation rate needed to initiate inner core freezing is where, on average, two critically sized nuclei can be expected to form within the supercooled duration and volume. For each of these nuclei, there is a 50% chance of remelting or freezing all supercooled liquids in the system. To produce a critical nucleation event in this waiting time, the $x^C = 0.05$ and $x^C = 0.1$ cases require $\delta T = 580^{+97}_{-71}$ K and $\delta T = 481^{+95}_{-67}$ K, respectively.

## Discussion

Through molecular dynamics simulations of supercooled Fe−C liquids, we show that elevated $x^C$ can significantly reduce the supercooling requirement for nucleation to spontaneously occur. This expands upon our previous study, which confirmed a similar but weaker effect at lower $x^C$ and narrows the gap between the supercooling required for inner core nucleation and the supercooling which is compatible with observations of the deep Earth. Our results for a carbon concentration of $x^C = 0.1$ do, strictly, represent a route to homogeneously nucleating solids at the centre of the completely liquid core because the allowable supercooling of the core ($\delta T = 420$ K) and the required supercooling for nucleation match within uncertainty (Fig. 3). The value of 420 K for the allowable supercooling is a maximum obtained by considering many different published melting curves and core temperature profiles to assess how much supercooling is compatible with the present-day thermal structure of the core[33]. Whilst this maximum value is not strictly self-consistent with the compositions explored here, the profiles of melting temperature and core temperature are relatively insensitive to composition, meaning that whilst the absolute values of $T_m$ will vary with composition, the maximum value of $\delta T$ can be the result of many different core chemistries. However, the maximum supercooling which is compatible with a more strict consideration of all available geophysical constraints is less than 100 K[35] (additional details are provided in the Supplementary Information). Despite this, it is of interest to understand how the required supercooling for inner core nucleation can be further reduced below this maximum value of 420 K to more plausible values through larger (>10 mol%) values of $x^C$.

The EAM developed in this study and used to define molecular dynamic simulations, which characterise nucleation behaviour of $Fe_{1-x}C_x$ alloys, cannot be used for $x^C$ above 0.1 (see "Methods"). We therefore extrapolate our results at lower $x^C$ (lower panel, Fig. 3) to predict how the supercooling requirement to spontaneously freeze the Earth's inner core might change with higher $x^C$. Previous studies suggest that up to 15.2 mol% (4 wt%) C might have entered Earth's core following accretion[37]. If extrapolated linearly to this concentration,

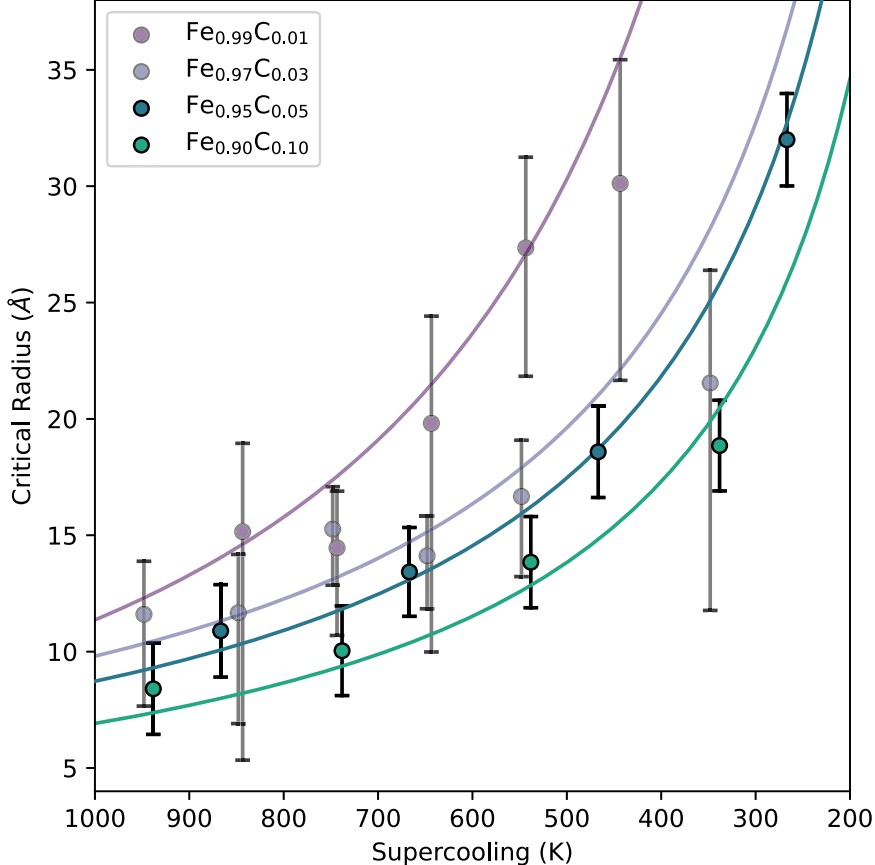

**Fig. 2 | Critical radii for nucleation in supercooled Fe–C alloys.** Critical radii for liquid $Fe_{1-x}C_x$ alloys between $x^C = 0.01$ and $x^C = 0.1$ ($x^C = 0.01$ and $x^C = 0.03$ cases are taken from Wilson et al.[33]), all at 360 GPa. $r_c$ is estimated from distributions of sub- critical nuclei. Uncertainties in $r_c$ are determined from the distributions of nuclei size at constant temperature (see Wilson et al.[31] for details). Temperature is shown as supercooling ($\delta T = T_m - T$, where $T_m$ is unique for each $x^C$, Fig. 1).

given a waiting time of $3.1 \times 10^{34}$ s m$^3$, inner core nucleation requires only 266 K of supercooling.

The melting temperature, $T_m$, at ICB conditions for a liquid carbon concentration $x^C = 0.1$ is around 5000 K (Fig. 1). This value is lower than the range 5300–5900 K obtained by previous studies for the Fe–O system with O concentrations in the range 8–17 mol%[38], though it is comparable to estimates of $T_m$ when H is a primary light element in the core[39]. The corresponding CMB temperature, estimated by projecting an adiabat from the ICB temperature using values from the Preliminary Reference Earth Model[25] and a Grüneisen parameter in the range 1–1.5[40] is ~3500 K, which is below estimates of the lower mantle solidus[41,42] as required by the observed absence of pervasive melt in the lower mantle.

Until now, we have assumed that the inner core nucleated homogeneously. Heterogeneous nucleation offers an alternate route to inner core formation but requires identification of a pre-existing solid surface to act as a nucleation site. Whilst nucleation in nature typically occurs in the presence of such surfaces, this still requires supercooling. In heterogeneous nucleation, the free-energy of homogeneous nucleation $\Delta G$ (see "Methods") is reduced due to a smaller solid-liquid interface being established compared to the homogeneous case (see Wilson et al.[35] for a review). The wetting angle between the nucleating metallic phase and the pre-existing solid controls the surface contact between the two solid phases and therefore defines the energetic benefit of heterogeneous nucleation compared to the homogeneous case.

One candidate heterogeneous nucleation site is oxides originating from the CMB, for example, precipitates from the cooling liquid core[43,44]. Previously considered oxides (FeO, MgO, SiO$_2$)[29,35] are not viable because

the do not have: sufficient density to be able to reach the centre of the Earth, where the core is first and most supercooled, and low solubility and high melting temperature in order to avoid dissolution or melting and remain solid in the core[35]. Even with these characteristics, the wetting angle between metals and oxides at 1 bar ranges from 110 to 180°, which corresponds to a $T_m$ reduction of at most 200 K for pure Fe in the Earth's core[29]. The resulting $\delta T$ remains incompatible with geophysical observations of the size of the inner core, meaning that a system with a smaller $\delta T$ for homogeneous nucleation of pure Fe is needed for a viable heterogeneous mechanism.

Metallic phases[29] typically have higher density and lower wetting angle compared to oxides. Identifying a phase that avoids dissolution and melting in liquid iron remains a challenge, and metals considered thus far are unlikely to reach Earth's centre[29,35]. At present, there is no material known to possess the required attributes to act as a site for heterogeneous inner core nucleation, and no geophysical scenario to explain how this material was delivered to the core. In the event that such a solid is discovered and required for inner core nucleation, the composition of the pre-existing solid itself will place a constraint on the bulk core composition, as will the nucleating phase.

The composition of Earth's core is likely to be more complex than the simple binary alloys we have considered[45,46]. However, it is nevertheless useful to discuss our simplified Fe–C compositions in the context of the available constraints. Geophysical constraints employ the radially varying core density and seismic wave speeds. C and O partition strongly into liquid iron on freezing[45,47] and are currently the primary candidates to explain the density jump $\delta\rho$ at the ICB. The C concentrations we consider are compatible with the values of the $\delta\rho = 0.6$–1.0 g cm$^{-3}$ derived from seismic normal modes[48], though

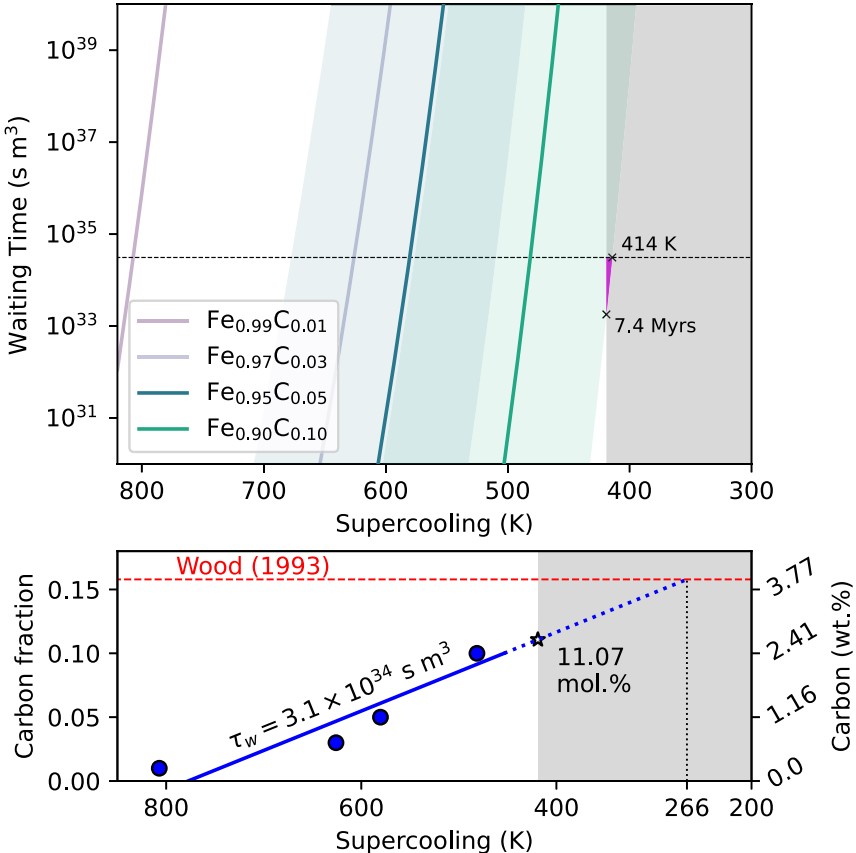

**Fig. 3 | Waiting times to observe inner core nucleation for Fe−C alloys.** Upper: Waiting time for a critical nucleation event to occur for four $Fe_{1-x}C_x$ compositions at a range of supercooling. Uncertainties are shown as shaded colours. Estimates from Wilson et al.[33] for $x^C = 0.01$ and $x^C = 0.03$ are also shown (without uncertainty for clarity). The dashed black line shows the maximum waiting time for an inner core half its present radius supercooled for 1 Gyr, and the grey shaded region represents supercooling values compatible with the present-day size of the inner core[33]. The pink area highlights areas of the $Fe_{0.9}C_{0.1}$ uncertainty envelop which represent a supercooling that is compatible with geophysically observations[33] Lower: Interpolation (solid blue line) and extrapolation (dashed blue line) of $Fe_{1-x}C_x$ results (solid lines, without exploring uncertainty) at $\tau_w = 3.1 \times 10^{34}$ s m³ to estimate the $\delta T$ needed to nucleate the inner core for values of $x^C$ up to the maximum proposed C content of the core, $x^C = 0.156$ (red line)[37].

plausible O concentrations can also explain the $\delta\rho$ observations. Matching the core mass as well as $\delta\rho$ requires at least one other light element that partitions more evenly between inner and outer core, e.g. S, Si[45] or H[49]. Ab initio calculations[39] indicate that Fe−C alloys with >15 mol% C, compatible with the geophysically allowable supercooling of the core, can match the seismically observed CMB and ICB density as well as the CMB P-wave velocity. However, the predicted ICB P-wave velocity is higher than observed. In the inner core, the anomalously high compressional/shear wave velocity is thought to relate to the presence of small amounts of O or C[16]. Depletion of the bulk silicate Earth compared to CI chondrites suggests that up to 15 mol% C could be in the core[37]. Recent experimental determinations of C partitioning between liquid iron alloys and silicate melts conducted in the pressure-temperature ranges 37−59 GPa and 4200−5200 K[50] and 49−71 GPa and 3600−4000 K[51] show that C becomes less siderophile as P and T increase, which, when applied to a specific model of core formation indicate that C does not partition strongly into the core. However, these estimates depend on mantle chemistry and the assumed core formation scenario, both of which are uncertain at present[16].

In summary, Fe−C binary alloys can satisfy some but not all constraints on the core composition. Studying nucleation is challenging even in binary alloys[30,33], and to date, no studies of ternary alloys have been attempted. Nucleation in the Fe−H system has also not been studied, though the weaker effect of H on the melting point[49] may suggest longer waiting times than we have found in the Fe−C

system. Other light elements have been shown to stabilise phases of iron[52], which might nucleate more readily than those favoured in the core[32]. However, this effect has not been observed in binary systems so far[33]. Ultimately, while many candidate compositions are able to reproduce available constraints from cosmochemistry[37], core formation[53], seismology[39,45,47], and Ni in the core may promote inner core nucleation[34], only an Fe−C composition has so far been shown to explain the nucleation of the inner core whilst satisfying the seismological requirement for light elements in the core. Hence, we argue that the process of inner core nucleation can provide a strong constraint on core composition, especially when combined with thermal evolution modelling[35] to incorporate further constraints. It is therefore worthwhile to reconsider previous inferences of core composition in light of this additional constraint. Future research should consider core compositions featuring C and Ni, as these elements might trigger inner core nucleation at geophysically compatible supercooling whilst satisfying seismological requirements for light elements.

The existence of a precipitated solid phase in the liquid core would present a route for the heterogeneous nucleation of the inner core. However, the required $\delta T$ for this mechanism will depend on the solid and the composition of the core. Our results have shown that only some compositions reduce $\delta T$. Therefore, a viable heterogeneous nucleation site, which avoids dissolution and melting, still places constraints on the core's composition.

Inner core nucleation subject to a supercooling of 200–400 K has potentially significant implications for interpreting the structure, dynamics, and evolution of Earth's core. The predicted supercooling would delay the inner core formation age predicted by core evolution models by $O(100)$ Myrs[54,55]. In classical evolution models with high core conductivity[56] this delay would likely imply a lack of power available to the dynamo prior to inner core formation, in conflict with paleomagnetic observations[57,58]. This observation lends support to evolutionary scenarios that include long-lived dynamo power supplied by precipitation of oxides at the CMB[59–63], though the effect of C on the partitioning behaviour at the CMB has not been systematically evaluated and may influence the power provided by precipitation. Sudden rapid growth of the inner core following nucleation may leave a signature in the paleomagnetic record owing to the additional latent heat and gravitational power to the dynamo[29], though the expected influence on field intensity and variability has not yet been studied in detail. Finally, delayed inner core formation may influence texturing of the inner core, for example by trapping liquids in the solid[54], and has been correlated with the existence of the innermost inner core[55].

## Methods

### Interatomic potential

We use CMD simulations of liquid $Fe_{1-x}C_x$ to characterise nucleation behaviour at a range of $T$ and $x^C$. To describe the interatomic forces and system energies in our simulations, we develop an EAM that is trained on ab initio calculations. The model is fit to reproduce the positions, energy ($E$) and $P$ of snapshots from ab initio molecular dynamics (AIMD) calculations run using the VASP software package[64] with the projector augmented wave method[65] and the PW91 generalised gradient approximation functional[66]. Details of these calculations follow Wilson et al.[33], which shares some of the same AIMD data at low $x^C$ used for fitting the potential. The EAM potential is validated against a separate suite of AIMD snapshots to ensure that $E$ and $P$ are accurately reproduced. The root mean square of fluctuations in $E$ is determined to be 0.292 and 0.316 eV per cell at 5000 K for $Fe_{0.95}C_{0.05}$ and $Fe_{0.9}C_{0.1}$, respectively, far less than $k_B/T$ (0.431 eV). Reproduction of liquid structure is confirmed by comparison of radial distribution functions, where average positions of neighbouring atoms in CMD simulations are within 0.05 Å of AIMD simulations for all interactions and all volume $V$, $T$, $x^C$ conditions. Further comparison of this potential with AIMD validation data is provided in the Supplementary Information.

AIMD simulations are performed by melting systems of 128 atoms with different carbon fractions (close to 20, 10 and 5 mol%) at 10,000 K for 1 ps before equilibrating at a target $T$ (4000, 5000 and 6000 K) for 1 ps and evolving the system at the target $T$ for 30 ps. The simulation cell volume is tuned for each composition and target $T$ to achieve a $P$ of 360 GPa. From the final 30 ps of simulation time, configurations are selected at every 100 fs as data on which the EAM is trained. The total energy $E$ of a $Fe_{1-x}C_x$ system is defined by the EAM as the sum of contributions from all atomic interactions

$$E = \sum_{i_{Fe}} E_i^{Fe} + \sum_{i_C} E_i^C + \sum_{i_{FeC}} E_i^{FeC}. \tag{1}$$

Each interaction between atoms $i$ and $j$ contains repulsive $Q$ and embedded $F$ contributions. $Q$ depends on the interatomic distance $r_{ij}$, which also defines an electron density $\rho_{ij}$ on which $F$ depends. $E$ for each type of interaction is given by

$$E_i^{Fe} = Q_i^{Fe} + F^{Fe}(\rho_i^{Fe})$$
$$= \sum_{i<j} \epsilon^{Fe} \left( a^{Fe}/r_{i_{Fe}j_{Fe}} \right)^{n^{Fe}} - \epsilon^{Fe} \dot{C}^{Fe} \sqrt{\rho_i^{Fe}}, \tag{2}$$

$$E_i^C = Q_i^C + F^C(\rho_i^C)$$
$$= \sum_{i<j} \epsilon^C \left( a^C/r_{i_C j_C} \right)^{n^C} - \epsilon^C \dot{C}^C \sqrt{\rho_i^C}, \tag{3}$$

$$E_i^{FeC} = Q_i^{FeC}$$
$$= \frac{1}{2} \sum_{i_{Fe} \neq j_C} \epsilon^{FeC} \left( a^{FeC}/r_{i_{Fe}j_C} \right)^{n^{FeC}}, \tag{4}$$

where the respective densities are

$$\rho_i^{Fe} = \sum_{j_{Fe} \neq i_{Fe}} \left( a^{Fe}/r_{i_{Fe}j_{Fe}} \right)^{m^{Fe}} + \sum_{j_C} \left( a^{FeC}/r_{i_{Fe}j_C} \right)^{m_{FeC}} \tag{5}$$

and

$$\rho_i^C = \sum_{j_C \neq i_C} \left( a^C/r_{i_C j_C} \right)^{m^C} + \sum_{j_{Fe}} \left( a^{FeC}/r_{i_C j_{Fe}} \right)^{m_{FeC}}. \tag{6}$$

Here, $\epsilon$, $a$, $n$, $m$ and $\dot{C}$ are free parameters to be fit for each interaction and are reported in Table 2. The primary difference to the parameters found in our previous study[33] is a reduction in $\epsilon^{FeC}$ and $\epsilon^C$, as well as $a^{FeC}$ and $a^C$.

### Melting temperatures

The melting temperatures of $Fe_{1-x}C_x$ are calculated with coexistence simulations using the EAM potential and the LAMMPS simulation package[67]. Systems of 128000 atoms are arranged into a long periodic cell where the $x$-axis is 3 times the length of the $y$ and $z$ axes. All atoms are initially arranged in a hexagonally close-packed structure with C atoms randomly replacing Fe atoms to achieve the desired concentration. This substitutional model is chosen based on ab initio evidence that C substitutions produce a lower free-energy state than interstitial C[47]. The positions of atoms in the central 50% of the simulation are initially fixed in space, whilst the other half is melted at 10,000 K for 5 ps. This procedure establishes the two-phase system. The entire system is then evolved at a target $T$ under the NVT ensemble, where the number of atoms, volume and temperature are held constant, for 1 ps to establish the target average kinetic energy. Finally, the system is evolved for 10 ps under the NVE ensemble (constant Number of atoms, Volume of the system and Energy of the system), allowing the solid region of the system to grow or melt. This process is repeated a minimum of 50 times for each composition, temperature and volume initial condition (where the volume is chosen such that the pressure of the system is 360(±2) GPa at the conditions of interest). As a result, a wide variety of configurations are sampled.

Once a system has reached equilibrium, the $T$ will lie on the melting curve, meaning that the time-averaged $T$ and $P$ provide a single $T_m$. The random distribution of C in the initial system provides many different initial $x^C$ for the solid and freezing and melting of the solid allow for C partitioning between the solid and the liquid. Systems with $x^C > 0.05$ in the solid see much of the solid melt before freezing a lower $x^C$ solid. This process shows that whilst C cannot diffuse freely in the solid over the timescale of these simulations, systems tend towards chemical equilibrium through freezing and melting. We note that we do not observe evidence of superionicity in these simulations, in line with previous ab initio studies[47], although this behaviour has been reported elsewhere, C-bearing Fe alloys at core conditions[68]. Simulations where $T$ and $P$ are stable on a picosecond timescale have $k_D = 4 \pm 2$ (where $k_D = \frac{x_{liquid}^C}{x_{solid}^C}$), which is consistent with ab initio calculations[47]. We estimate the uncertainties of each $T_m(v, x)$ point from the fluctuations of $T$ and $P$ over the final 1 ps of simulation time

**Table 2 | Parameters for EAM model fit to FPMD data at several C concentrations and temperatures**

|      | $\epsilon$     | $a$          | $n$        | $m$        | $\dot{c}$   |
|------|----------------|--------------|------------|------------|-------------|
| Fe   | 0.166200 eV    | 3.471400 Å   | 5.930000   | 4.788000   | 16.550000   |
| FeC  | 0.384726 eV    | 2.601660 Å   | 4.380769   | 4.933012   |             |
| C    | 0.019805 eV    | 2.311113 Å   | 9.532860   | 6.967342   | 13.880981   |

Fe values, from Alfè et al.[36], are fixed during fitting.

and discard any simulations that entirely freeze, melt, or do not achieve equilibrium. Because of the constant volume and energy conditions, $T$ and $P$ are unknown prior to the simulation setup. In order to define $T_m(P, x)$, we explore a range of initial $T$ and $v$ and interpolate our results for $T_m(360\,\text{GPa}, x)$.

**Nucleation theory**

In this study, we use CNT to describe the nucleation behaviour of CMD simulations of iron alloy liquids at the conditions of Earth's core. Previous studies[30–33] have routinely found that predictions from CNT are consistent with outputs from MD simulations, accurately describing the distribution of nucleus sizes and the dependence of nucleation rate on supercooling. We note that our CMD simulations are completely independent of CNT; indeed, these simulations have been used to show that non-classical effects such as pressure waves have no effect on the nucleation of solids in Earth's core[30]. According to CNT, the requirement for liquids to be supercooled prior to freezing via homogeneous nucleation arises from a competition between two energetic contributions to the total free energy, $\Delta G$, associated with forming a solid nucleus in a supercooled liquid. The first contribution is the free-energy release, $g^{sl}$, associated with transforming supercooled liquid into a solid, which is always favourable when below the melting temperature and occurs through random fluctuations in the liquid, producing 'solid-like' configurations of atoms. The second contribution, $\gamma$, is associated with forming an interface between the liquid and solid and is always unfavourable. These two components are scaled by the volume and surface area of the newly formed nucleus of radius $r$ to define a total free-energy change on formation

$$\Delta G(r) = \frac{4}{3}\pi r^3 g^{sl} + 4\pi r^2 \gamma \tag{7}$$

for spherical particles.

The rate $I$ at which a nucleus of radius $r$ forms is defined by Boltzmann statistics:

$$I(r) = I_0 \exp\left(\frac{-\Delta G(r)}{k_B T}\right), \tag{8}$$

where $k_B$ is Boltzmann's constant and $I_0$ scales the nucleation rate of the specific system. Eq. (8) shows that small nuclei are likely to form often (or equivalently, require less waiting time ($\tau_w \approx I^{-1}$) before they occur). However, Eq. (7) shows that these nuclei will remelt rather than grow because of the large influence of surface area on the free energy at small $r$. Despite a low probability, continued growth is possible given a sufficiently long waiting time and a large system volume to observe random fluctuations that produce a larger nucleus. Above a critical radius $r_c = -2\gamma/g^{sl}$ at the peak of $\Delta G$, the volume term in Eq. (7) increases with radius faster than the surface term, meaning that whilst still having an overall unfavourable free energy for forming a nucleus, continued growth is thermodynamically favoured when compared to remelting. Greater supercooling requires a smaller $r_c$ in order to freeze a system, which in turn requires less waiting time for the critical event to spontaneously occur.

The rate at which a nucleus of radius $r$ spontaneously forms in a supercooled liquid is given by Eq. (8). When framed in terms of $r_c$, the inverse of nucleation rate describes the average duration before a supercooled system will experience a critical nucleation event and freeze

$$\tau_w = \tau_0 \exp\left(\frac{\Delta G(r_c)}{k_B T}\right), \tag{9}$$

where

$$r_c = \frac{-2\gamma}{g^{sl}}. \tag{10}$$

The prefactor $\tau_0$ can be described by

$$\tau_0 = \frac{z}{NS}, \tag{11}$$

where the Zeldovich factor $z$ is related to $g^{sl}$ through

$$z = \left(\frac{\frac{4}{3}\pi r_c^3 g^{sl}}{k_B T}\right)^{-1/2}. \tag{12}$$

and $N$ and $S$ are the number of available nucleation sites and the rate of nuclei growth, respectively. To quantify $N$, $S$ and $I(r)$, solid-like arrangements of atoms are identified at each timestep in the same manner as our previous studies[31,33] following Rein ten Wolde et al.[69]. Therefore, all quantities required to calculate $\tau_w$ are accessible via CMD calculations. Because $r_c$ is predicted to be large for the $P$ and $T$ of the early Earth's supercooled liquid core[31], simulations at >4000 K and 360 GPa cannot be expected to produce a nucleus of the critical size (>30 Å). Instead, $r_c$ is predicted by recording the rate at which smaller nuclei (which are more common) are observed in simulations, informing $I_T(r)$ where $r$ is small. At a fixed $T$ all quantities in Eq. (7) are constant, so we can write

$$-\ln(I_T(r)) \propto \Delta G_T(r) \tag{13}$$

and the distribution of nuclei observed in simulations describes the form of $\Delta G_T(r)$ but not the amplitude. Nuclei are observed to be approximately spherical for $r > 2$ Å, in line with our previous studies[31,33]. The radius of any recorded nuclei is taken as $r = (v/(\frac{4}{3}\pi))^{\frac{1}{3}}$, where $v$ is the volume occupied by a nucleus. The form of the free-energy barrier can be represented by

$$\Delta G_T(r) = 4/3\pi r^3 A + 4\pi r^2 B, \tag{14}$$

where $A$ and $B$ are variables at each $T$, meaning that $r_c$ can be estimated via $r_c = -2B/A$, equivalent to Eq. (10). When fitting for $A$ and $B$, only $I_T(\geq 2\,\text{Å})$ is considered to avoid errors associated with small non-spherical nuclei. If repeated for a range of $T$ (and therefore $\delta T$) $r_c(T)$ is obtained. The free parameters $\gamma$, $h_f$ and $h_c$ are then found by fitting for $r_c(T)$ through

$$r_c(T) = \frac{-2\gamma}{h_f \frac{\delta T}{T_m}(1 - h_c \delta T)}, \tag{15}$$

where the $h_f$ is the enthalpy of fusion and $h_c$ accounts for non-linearity with temperature when defining the free energy liberated by freezing supercooled liquid

$$g^{sl} = h_f \frac{\delta T}{T}(1 - h_c \delta T). \tag{16}$$

This representation of $g^{sl}$ does not consider $x$, meaning that $h_f$ and $h_c$ must be unique to each composition. In a previous study[70] we considered $g_{sl}(T, P, x)$ including ideal mixing $g_{sl}$ and found that, for an Fe–O system, the mixing effect is negligible. In the Fe–C systems of this study ideal mixing contributions ($g_{mix} = R_g T \ln x$, where $R_g$ is the gas constant) are also small compared to $g_{sl}$ and non-ideal effects have been found to be minor[39].

## Data availability
The molecular dynamic data generated in this study have been deposited in a Zenodo database [https://doi.org/10.5281/zenodo.15310896].

## Code availability
The LAMMPS (Large-scale Atomic/Molecular Massively Parallel Simulator) package is provided freely at https://lammps.sandia.gov/. The code used for analysis in this study is provided in the Zenodo repository alongside data and is otherwise freely available.

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

## Acknowledgements

We gratefully acknowledge Natural Environment Research Council (NERC) grant, reference NE/T000228/1, which supports all authors on this project. A.J.W. and C.D. acknowledge support from the NERC grant NE/V010867/1. A.M.W. and C.D. acknowledge support from the NERC grant NE/T004835/1. D.A. acknowledges support from the NERC grants NE/M000990/1 and NE/R000425/1. We also thank the reviewers for their helpful contributions.

## Author contributions

A.J.W., C.J.D., A.M.W. and D.A. jointly conceived of the study; A.J.W. performed research and analysis; and A.J.W., C.J.D. and A.M.W. co-wrote the paper.

## Competing interests

The authors declare no competing interests.
