## [Transparent Peer Review file · Nature Communications]

Constraining Earth's core composition from inner core nucleation

Corresponding Author: Dr Alfred Wilson

Version 0:

Reviewer comments:

Reviewer #1

(Remarks to the Author)

Wilson et al. investigate Earth's core composition by studying inner core nucleation. The problem originates from the "inner core nucleation paradox" proposed in previous work. By testing different binary Fe alloys in their earlier study (Ref. 33), the authors found Fe-C to be a possible candidate for the inner core. Based on this observation, they refined their Fe-C interatomic potential and discovered that a carbon content of ~ 0.1 could lead to reasonable supercooling, which is required to nucleate the initial solid phase of the inner core. In this way, they offer a new approach to constraining the core's light element content, independent of previous constraints from seismology and thermodynamics.

The idea of using initial nucleation to constrain the light element content is novel. I believe this work is worthy of publication in a high-impact journal for a broad audience. However, the current manuscript has a few issues that need to be addressed to make this approach more compelling and robust. Below are some suggestions.

1. In the T_m calculations, the authors describe constructing the initial lattice by replacing Fe with C, implying a substitutional model. However, in a substitutional model, direct MD simulations may not yield a realistic melting point because C atoms remain nearly fixed in lattice sites. While the authors use a few snapshots to estimate the melting point, this approach is not well established. Monte Carlo simulations are typically required to equilibrate the solute distributions in such a solid solution.
2. Carbon has been suggested to be superionic in the inner core (He et al. 2022). However, the authors seem to have overlooked this work when computing the partition coefficient.
3. The authors modify their previous Fe-C interatomic potential. These modifications and their impact should be discussed. Comparisons of the liquid radial distribution function and scatter plots with DFT data for energy and forces should be presented.
4. To my understanding, the current potential is still not applicable for x_C larger than 0.1. Can the authors discuss possible reasons for this limitation?
5. In the nucleation simulations, the authors assume a spherical nucleus. However, deviations from spherical shape are commonly encountered across a wide range of systems. Such deviations could introduce errors in the determination of nucleation rates, an issue that has been discussed in detail in the materials science community (e.g., J. Chem. Phys. 155, 040901 (2021)). If the manuscript is to be published in a multidisciplinary journal, I suggest the authors clarify this assumption. Additionally, they should provide a clear definition of how they determine the radius of the nucleus when it is not perfectly spherical.
6. Equations 16–17 were derived for pure elemental systems. It is unclear whether they remain valid for binary systems, given that the equilibrium chemical composition changes between the bulk solid and liquid phases. Moreover, it is uncertain whether the nucleus maintains the same chemical composition as the bulk phase and whether the partition coefficient remains constant with the temperature. These factors could significantly affect $g^{\{sl\}}$, which in turn influences the nucleation rate exponentially. The authors need to carefully investigate these effects.
7. The authors do not provide much analysis of the as-nucleated phase, which, in my opinion, is a crucial outcome

representing the initial form of the inner core's solid. What crystalline phase forms? What is the nature of C in the Fe lattice—substitutional, interstitial, dimeric, or superionic? The spontaneously formed phase is vital for interpreting the results.

8. This study and Ref. 33 focus on light elements. However, nickel and other heavy elements in the core may also influence initial nucleation. The authors should include some discussion on these potential effects.

Reviewer #2

(Remarks to the Author)

This paper presents new data on the influence of carbon (at high concentrations) on the supercooling required for homogeneous nucleation of Earth's inner core. The authors find that C has a larger effect on the required supercooling than other individual "light elements" that have been considered to date. This is a valuable finding that deserves to be published. The required supercooling (hundreds of K) is still too large to plausibly explain nucleation of the inner core, but it is a step in the right direction.

Detailed Comments

39-40 Cooling of the core is determined by heat transfer across the core-mantle boundary, not by the thermal conductivity of the core.

156-62 A little more discussion of this waiting time would be helpful. A large fraction of the present inner core volume sitting for 1 Gyr at a constant (large) supercooling is unrealistic and hence does not appear to be a "moderate" assumption. On the other hand, Fig. 3 shows that the required supercooling is fairly insensitive to the exact value of the waiting time.

164-7. This is really pushing it. Even if the required supercooling at 10% C is just within the maximal bound on the core temperature change from nucleation to the present, it would require crystallization of essentially the entire inner core from a supercooled state, which would have occurred very rapidly, essentially yesterday, and with the "inner" inner core (if it represents early, rapid crystallization from a supercooled state) representing essentially the entire core. This is not a plausible scenario. The same goes, really, for the lower supercooling values discussed below, at higher C concentrations. In my opinion, the paper will be more effective if it's up front about the plausible limits and doesn't overstretch the argument for C.

252. Certainly a novel constraint, with the potential to be strong. But there are still many uncertainties, including whether there is a plausible route via heterogeneous nucleation on a metallic substrate.

263-78. The Conclusion section consists mainly of points that are outside what is discussed in the main body of the paper, and not particularly dependent on the results presented.

Reviewer #3

(Remarks to the Author)

The manuscript builds on previous research investigating the supercooling behavior of core compositions relevant to Earth's core, specifically focusing on the binary Fe-C system. The authors employ molecular dynamics simulations to determine key parameters governing the transition from a fully liquid state to the onset of freezing. They then apply classical nucleation theory to interpret their findings and derive critical parameters, such as the required supercooling and the critical nucleus size.

The methodology, largely adopted from their previous work, has been well established in earlier studies. In this paper, they extend the approach to additional compositions within the Fe-C system, bringing their analysis closer to the supercooling conditions constrained by geophysical models. The central contribution of the manuscript, as I understand it, is to introduce a paradigm shift—positioning the study of supercooling behavior as an independent yet complementary constraint in planetary core research.

While the paper does not introduce significant methodological innovations (at least not entirely clear from manuscript), I recognize the authors' effort to highlight this new perspective. This work is timely, as the challenge of nucleating Earth's inner core was identified by Huguet et al. (2018) and has remained an open question since. The authors, through this and previous work, have steadily developed a novel and constructive approach to addressing this issue. In this context, the present manuscript represents a crucial step in establishing supercooling behavior as a key research focus in planetary science.

The paper is well-written, concise, and clear. In my opinion, it is suitable for publication. Below, we (ECR and me) provide some general remarks and minor comments that should be addressed before final acceptance.

General Comments:

- It is somewhat unfortunate that the simulations fail for $x > 0.1$, as these cases appear to be the most relevant in terms of geophysical constraints. It would be helpful to provide a more detailed explanation of the issues causing these failures. While there are hints throughout the text, a clearer and more concrete statement would be beneficial—does the problem stem from the limitations of the EAM potential? What are the implications of this failure?
- The discussion of geophysical constraints could be expanded. What specific geophysical models are referenced? To what extent are they independent of the assumed concentrations? Given that supercooling behavior is being proposed as a new constraint on core composition, a more detailed explanation of these models and their role in this context would strengthen the manuscript.
- The paper would benefit from a clearer distinction from the authors' previous work, particularly Wilson et al. (2023) [33]. It would be helpful to explicitly state any methodological differences and clarify how this study extends prior research (e.g., by applying the same methodology to mixtures with higher carbon concentrations). For instance, the direct reference to [33] in lines 99–101 could be expanded with a statement such as: "We extend previous studies by considering ..." to emphasize the novelty of the present work.
- Along the same lines: In what way do the contributions unique to this paper—such as the investigation of the F-C mixture at higher carbon concentrations — establish this admissibility criterion in a manner not previously considered?
- Similarly, the discussion of results could be strengthened with a summarizing statement. For example: "Through the simulations of the Fe-C mixture at higher x conducted in this study, we have taken a first step toward assessing the undercooling admissibility criterion ..."

Minor comments:

- l. 119: The definition of abbreviation EAM for "Embedded Atomic Model" is missing. It is only introduced in the later Sec. 5.
- l. 121: the variable x should probably be an x^C
- Fig. 1: in the lower plot, the y axis label is missing a variable name (e.g. x or x^C)
- l. 128: the nucleation rate $I(r)$ is used as a function. Since it was only mentioned once before repeating at least the name here would make it easier to read
- l. 137: starting a sentence with a variable τ_w . Change to something like "The average waiting time τ_w is computed as $\tau_w = \tau_0 \exp(\dots)$ "
- l. 145: it is not easily traceable for me what the differences of the interatomic potential are wrt. to the existing work from Wilson et al. 2023. The methodology described in 5.1 seems to be exactly the same. For me, a highlight of differences would be useful to assess in which regard this work is new.
- l. 148: missing "break" between symbols ΔT and r_c , recommend to change to "at large ΔT critical radii r_c for all ..."
- ll. 148-150: please elaborate a bit on the type of instability / why the method breaks down (see general comment above). It is fair the method has limitations but it would be helpful to understand them better
- l. 157: Waiting time is given in time/volume. It would be great to have one or two explanatory sentences as to how this relates to a time in the classical sense.
- l. 328: the abbreviation "NVE" has not been defined
- l. 414: please fix the link to the zenodo repository to include a full copyable doi link, e.g. www.doi.org/10.5281/zenodo.13144422 . Also maybe rephrase the sentence such it does not end with the link to avoid merging the sentence-ending dot with the URL

Small comments (mostly writing):

- l.66: delete "result in ..." (duplication)
- l. 79: missing comma after "Here"
- l. 94: "Ref" shown before reference [34]
- l. 101: missing space before reference [33]
- l. 103: potential comma after "nucleation"
- ll. 129-130: repeated word "which"
- l. 147: missing comma before "although"
- Table 1: maybe put the unit of variables in its own column in the tabular to fix alignment
- l. 237: missing comma -> "which, when applied to a specific model of core formation, ..."
- ll. 238-239: repeated use of the word "strongly", suggest to replace once
- ll. 346-347: double use of the word "consistent"
- l. 422: Section heading weirdly transitions into the following paragraph.

Reviewer #4

(Remarks to the Author)

Version 1:

Reviewer comments:

Reviewer #1

(Remarks to the Author)

The authors have made efforts to address most of the reviewer's concerns, resulting in substantial improvements to the manuscript. However, they cannot examine the critical nucleus, as their technique focuses solely on sub-critical configurations. This approximation may overlook important pathways for solidification.

Nevertheless, the present results are significant enough for core studies. I think the paper can be published.

Reviewer #2

(Remarks to the Author)

In my opinion, the authors have addressed effectively the points made by the reviewers, and the article is suitable for publication.

Reviewer #3

(Remarks to the Author)

The reviewers carefully addressed all points raised. The manuscript is now - in my opinion - ready for publication. One remark: The answer of the authors to one point raised by reviewer number 2 is not fully correct and I just wanted to clarify:

Reviewer 2: 39-40 Cooling of the core is determined by heat transfer across the core-mantle boundary, not by the thermal conductivity of the core.

Authors: The reviewer is correct. Mantle convection sets the heat flow across the core mantle boundary and the thermal conductivity of the core controls the adiabatic heat flow through the core. The thermal conductivity of the core controls the convective vigour of the core. I.e. a high core conductivity will result in more vigorous convection than a low core conductivity, if the core-mantle boundary heat flux is the same in both cases.
We have modified the text at lines 38-39 to make this statement clear.

High core conductivities will hamper convection as most of the heat can be transported by heat conduction. However, the way the authors addressed this in the manuscript is fine as it is only a general statement ("...thermal conductivity k , which plays a vital role in determining the vigour of convection in the liquid core...").

Reviewer #4

(Remarks to the Author)

Response to referees

Original reviewer comments: Italics

Authors' responses: regular font

Location of changes in revised manuscript: bold

Reviewer #1 (Remarks to the Author):

Wilson et al. investigate Earth's core composition by studying inner core nucleation. The problem originates from the "inner core nucleation paradox" proposed in previous work. By testing different binary Fe alloys in their earlier study (Ref. 33), the authors found Fe-C to be a possible candidate for the inner core. Based on this observation, they refined their Fe-C interatomic potential and discovered that a carbon content of ~0.1 could lead to reasonable supercooling, which is required to nucleate the initial solid phase of the inner core. In this way, they offer a new approach to constraining the core's light element content, independent of previous constraints from seismology and thermodynamics.

The idea of using initial nucleation to constrain the light element content is novel. I believe this work is worthy of publication in a high-impact journal for a broad audience. However, the current manuscript has a few issues that need to be addressed to make this approach more compelling and robust. Below are some suggestions.

We thank the reviewer for their encouraging assessment of our study, its novelty and the potential for impact. The comments made are helpful for improving the clarity of our approach. We have addressed the reviewer's comments below.

1. In the T_m calculations, the authors describe constructing the initial lattice by replacing Fe with C, implying a substitutional model. However, in a substitutional model, direct MD simulations may not yield a realistic melting point because C atoms remain nearly fixed in lattice sites. While the authors use a few snapshots to estimate the melting point, this approach is not well established. Monte Carlo simulations are typically required to equilibrate the solute distributions in such a solid solution.

The reviewer is of course correct that care must be taken when calculating melting temperatures through this two-phase coexistence method. We have taken the necessary precautions to ensure that our results are robust. We define the composition of our two-phase coexistence calculations by substituting a random selection of Fe atoms with C and where these atoms are within the solid region they are unable to diffuse within the timespan of these simulations. This choice of a substitutional model is based on the study by Li et al. (2019, <https://doi.org/10.1029/2019JB018789>) where substitutions are found to be lower free-energy configurations than interstitial ones, implying that this is how C would be held in the Fe crystal lattice at core conditions. To ensure that we sample configurational space sufficiently and produce accurate melting temperatures we conduct at least 50 simulations per temperature, volume, composition condition, each with a unique initial distribution of carbon. Many of these two-phase simulations see a large degree of melting and refreezing as the phase boundaries move within the simulation volume during the equilibration performed in every simulation. This means that C atoms are released to the melt and sometimes trapped in the solid, although the solid predominantly forms with little C upon refreezing, in agreement with the partitioning results of Li et al. (2019). These cases show that C atoms are mobile in these simulations and that equilibrium conditions can be met even without diffusion. Ultimately, this is a robust method for calculating the melting temperatures of binary systems using molecular dynamics simulations, as proven through application to a number of high pressure and temperature systems (for example Fe-S, Fe-Si,

Fe-O, Alfe et al., 2002 <https://doi.org/10.1063/1.1464121>; Li-H, Ogitsu et al., 2003 <https://doi.org/10.1103/PhysRevLett.91.175502>; MgO, Belonoshko et al., 2010, <https://doi.org/10.1103/PhysRevB.81.054110>; MgSiO₃, Deng et al., 2023 <https://doi.org/10.1103/PhysRevB.107.064103>).

We have added text in the Methods section on the choice of a substitutional model (lines 360-361), the sampling of configurational space (lines 367-371) and the diffusion of C and melting-refreezing of the solid (lines 377-379). We have also added text to lines 120-122 of the results section which briefly describes to the care taken on these matters and direct the reader to the Methods section for greater detail.

2. Carbon has been suggested to be superionic in the inner core (He et al. 2022). However, the authors seem to have overlooked this work when computing the partition coefficient.

We thank the reviewer for highlighting this paper. The transport properties of C are not the focus of this study although we note that no evidence of C superionicity is seen in our simulations. This finding is in line with that of Li et al. (2019, <https://doi.org/10.1029/2019JB018789>) who do not find any diffusive behaviour in their ab initio simulations of Fe₆₀C₄ through root mean square displacement testing. We conclude that this topic is of great interest but not yet resolved and that the simulations presented here are not designed to interrogate the transport properties of the inner core, largely due to the out of equilibrium (supercooled) condition we are studying in these simulations.

To address this topic we have added text at lines 379-381 including reference to He et al. (2022) and Li et al. (2019).

3. The authors modify their previous Fe-C interatomic potential. These modifications and their impact should be discussed. Comparisons of the liquid radial distribution function and scatter plots with DFT data for energy and forces should be presented.

We have added detail on how the potential of our previous study (Wilson et al. 2023, <https://doi.org/10.1016/j.epsl.2023.118176>) differs from the potential presented here and have now included a supplementary information file which provides comparison of energy, pressure and radial distribution function between this new potential and independent ab initio data. This supplementary material demonstrates that our revised interatomic potential robustly reproduces the behaviour of ab initio calculations with 10 mol.% C.

Reference to this supplementary information is now made on lines 333-334 and the difference in the parameters of the potential compared to our previous study are noted on lines 352-353.

4. To my understanding, the current potential is still not applicable for x^C larger than 0.1. Can the authors discuss possible reasons for this limitation?

In our simulations at $x^C > 0.1$ we observe the liquid separate into distinct regions of high and low x^C . This apparent phase separation produces regions where x^C is approximately 0.3, far higher than the ab initio data used to train and validate the EAM potential. Therefore, we cannot validate this behaviour without producing an additional dataset of ab initio calculations where $x^C \gg 0.1$. This dataset would need to include x^C from 0.0 to at least 0.3, meaning more than doubling the size of our existing dataset. This is a considerable task and would require significant computational resources. Given that there is little expectation for the core to host such high concentrations of C (mostly due to limitations that seismological constraints place on the x^C of the core, see Hirose et al., 2021, <https://doi.org/10.1038/s43017-021-00203-6>, for a review) there is limited motivation for such investment so we instead chose to omit results for $x^C > 0.1$ from our study. Instead, we focus on how results for x^C up to 0.1 demonstrate that inner core nucleation will provide a constraint on core composition when a viable mechanism is identified.

We have added text to lines 158-162 of the results section to explain this behaviour and the reason that we omit the results from our study.

5. In the nucleation simulations, the authors assume a spherical nucleus. However, deviations from spherical shape are commonly encountered across a wide range of systems. Such deviations could introduce errors in the determination of nucleation rates, an issue that has been discussed in detail in the materials science community (e.g., J. Chem. Phys. 155, 040901 (2021)). If the manuscript is to be published in a multidisciplinary journal, I suggest the authors clarify this assumption. Additionally, they should provide a clear definition of how they determine the radius of the nucleus when it is not perfectly spherical.

The reviewer raises a valid point which we have taken great care to address in our previous work (Wilson et al. 2021, <https://doi.org/10.1103/PhysRevB.103.214113>; Wilson et al. 2023, <https://doi.org/10.1016/j.epsl.2023.118176>) and this study. We have clarified that our previous study on these supercooled liquids shows that for all but the smallest nuclei a spherical assumption is valid and represents the geometry of observed nuclei well. To avoid errors that non-spherical nuclei can introduce, we do not include nuclei smaller than 2 Å in our determination of critical radii.

The text is on lines 145, 440-442 and 445-446 has been modified for clarity on these details.

6. Equations 16–17 were derived for pure elemental systems. It is unclear whether they remain valid for binary systems, given that the equilibrium chemical composition changes between the bulk solid and liquid phases. Moreover, it is uncertain whether the nucleus maintains the same chemical composition as the bulk phase and whether the partition coefficient remains constant with the temperature. These factors could significantly affect g^{sl} , which in turn influences the nucleation rate exponentially. The authors need to carefully investigate these effects.

The reviewer is correct that the formulation of classical nucleation theory was originally intended for single component systems. In this study we simply use this formalism to describe the distribution of nuclei sizes in our results and find that it is adequate in doing so, as do our previous studies employing this approach (Wilson et al. 2021, <https://doi.org/10.1103/PhysRevB.103.214113>; Wilson et al. 2023, <https://doi.org/10.1016/j.epsl.2023.118176>, where the latter focused on two-component systems including Fe-C). Because we allow nucleation to spontaneously occur in these molecular dynamic systems, we do not make any assumption of the composition of the nucleating phase or partition coefficients. This means that by inferring g^{sl} from the observed distributions of nuclei in our simulations, we can learn about the behaviour g^{sl} with composition and supercooling. We find that g^{sl} is not greatly modified by x^{C} , in line with our prior study (Wilson et al. 2023, <https://doi.org/10.1016/j.epsl.2023.118176>), and that the difference in interfacial energies is predominantly responsible for the difference in system behaviours.

Additionally, we have recently published a study (Walker et al., 2025, <https://doi.org/10.1098/rspa.2024.0505>) which does consider the modification of g^{sl} for systems where solid and liquid compositions are expected to differ significantly. In this study the effect of the partitioning specifically is shown to be small, even in cases where it is maximised. If we take a similar approach for our study and evaluate the ideal mixing contribution to the free energy ($g_{\text{mix}} = R_g T \ln(x)$) we find that this term is at least four orders of magnitude smaller than the free energy difference between solid and liquid phases. This analysis assumes that non-ideal contributions are small, which is found by Umemoto and Hirose (2022, <https://doi.org/10.1016/j.epsl.2019.116009>) to be the case for several light elements in the core, including carbon.

We have added text on lines 131-132 and lines 138-141 to explain that classical nucleation theory has been shown to accurately describe the nucleation rate of these

systems in our previous work. In the Methods section, alongside equations 16 and 17 (lines 451-456), we explain that mixing terms, which are not included in classical nucleation theory, are expected to be negligible whilst explaining the assumptions made therein.

7. The authors do not provide much analysis of the as-nucleated phase, which, in my opinion, is a crucial outcome representing the initial form of the inner core's solid. What crystalline phase forms? What is the nature of C in the Fe lattice—substitutional, interstitial, dimeric, or superionic? The spontaneously formed phase is vital for interpreting the results.

The reviewer is correct that the initially nucleated phase of the inner core is of great interest for understanding first solids of the inner core, as well as its the subsequent growth and properties. However, our approach samples sub-critical nuclei, which in our simulations all go on to remelt. In our previous study (Wilson et al., 2021, <https://doi.org/10.1103/PhysRevB.103.214113>) we use the same technique alongside a “brute force” approach of applying extreme supercooling to trigger critical nucleation within timespans accessible to molecular dynamics simulations. In these cases, we find that the as-nucleated phase will be metastable or unstable, relaxing to a more stable state if observed for an extended duration (picoseconds after the nucleation event). This mechanism of “two-step” nucleation is also found by Sun et al. (2022, <https://doi.org/10.1073/pnas.2113059119>) where the body centred cubic structure of iron nucleates more readily than the more thermodynamically stable hexagonally close packed phase. Two-step nucleation therefore illustrates that the connection between sub-critical nuclei phases and the young inner core is complicated, especially on geological timescales where millions of years of annealing, diffusion and deformation have altered crystal phases and textures.

As detailed in the response to comment #6, the nature of the nucleating phase is not crucial to understanding the energetics and rates of nucleation in the systems we are studying. It is precisely these quantities which allow us to identify compositions which can trigger freezing at supercooling which is compatible with Earth's core. Therefore, whilst the details of the as-nucleated phase are crucial for understanding the properties of the inner core in its smallest and youngest state, these details do not change our demonstration that inner core nucleation can be a constraint on the composition of the core.

Our prior studies contain information on the as-nucleated phase which applies to this study as well (Wilson et al. 2021, <https://doi.org/10.1103/PhysRevB.103.214113>; Wilson et al. 2023, <https://doi.org/10.1016/j.epsl.2023.118176>). In Wilson et al. (2023) we find that the phase of sub-critical nuclei in supercooled Fe-C liquids are best described as defect rich hexagonally close packed with C being incorporated substitutionally, although the defects (often planar in nature) make interpretation of phase difficult.

We have added text on lines 145-147 to describe the observed phase of the sub-critical nuclei in this study. Here we also direct the reader to our prior studies which contain a more detailed investigation.

8. This study and Ref. 33 focus on light elements. However, nickel and other heavy elements in the core may also influence initial nucleation. The authors should include some discussion on these potential effects.

The reviewer is correct. The effect of metals in the core is likely to have an impact on the nucleation rates of these systems. We note that Sun et al. (2024, <https://doi.org/10.1073/pnas.2316477121>) found that Ni in particular may reduce the supercooling required to nucleate the inner core by ~400 K. In this study we focus exclusively on the effect of C because of the accelerated nucleation rate if Fe-C liquids discovered in our previous study. When combined, Ni and C may explain inner core nucleation with a plausible degree of supercooling ($dT \ll 400$ K). This would be significant because core thermal histories require that $dT < 100$ K (Wilson et al. 2025, <https://doi.org/10.1038/s43017-024-00639-6>), whilst seismology requires that the core is

host to light elements. A combination of Fe, Ni, C and another light element could therefore explain the thermal evolution of the deep Earth, inner core nucleation and seismological observations, which is the main message of this manuscript.

We have added text at lines 284-287 to note that outside of carbon and other light elements, metallic elements present in the core might play a part in the process of inner core nucleation and would be required to provide a complete model of core composition. We have also noted that Fe-Ni-C-X alloys make a compelling next step for investigation on line 290-293.

Reviewer #2 (Remarks to the Author):

This paper presents new data on the influence of carbon (at high concentrations) on the supercooling required for homogeneous nucleation of Earth's inner core. The authors find that C has a larger effect on the required supercooling than other individual "light elements" that have been considered to date. This is a valuable finding that deserves to be published. The required supercooling (hundreds of K) is still too large to plausibly explain nucleation of the inner core, but it is a step in the right direction.

We thank the reviewer for their careful consideration of the manuscript. We are pleased that they find the work to be valuable and that they support publication of the manuscript. The comments raised are valid and valuable, we have addressed each of them below.

Detailed Comments

39-40 Cooling of the core is determined by heat transfer across the core-mantle boundary, not by the thermal conductivity of the core.

The reviewer is correct. Mantle convection sets the heat flow across the core mantle boundary and the thermal conductivity of the core controls the adiabatic heat flow through the core. The thermal conductivity of the core controls the convective vigour of the core. I.e. a high core conductivity will result in more vigorous convection than a low core conductivity, if the core-mantle boundary heat flux is the same in both cases.

We have modified the text at lines 38-39 to make this statement clear.

156-62 A little more discussion of this waiting time would be helpful. A large fraction of the present inner core volume sitting for 1 Gyr at a constant (large) supercooling is unrealistic and hence does not appear to be a "moderate" assumption. On the other hand, Fig. 3 shows that the required supercooling is fairly insensitive to the exact value of the waiting time.

This is a valuable comment raised by reviewer 2 and 3. 420 K of supercooling and a waiting time of 1 Gyr are the maximum plausible values for these quantities given current knowledge. We referred to the "moderate" case as such because it represents a scenario when half of the present day inner core radius is supercooled for 1 Gyr and after inner core nucleation the remaining half of the inner core radius could have grown slowly. In this sense the case is moderate because it does not require the entire inner core to have frozen in the immediate past, which is what we would term the "extreme" case. Despite both of these cases seeming unlikely, our recent review on the topic (Wilson et al. 2025, <https://doi.org/10.1038/s43017-024-00639-6>) shows that there are currently no geophysical observations which directly prohibit an extended period of supercooling in the core. Because of the exponential relationship between supercooling and waiting time it makes little difference to the interpretation of our results if a moderate or extreme case is chosen, as the reviewer suggests. Following this comment, we feel that an improved description of the maximum allowable waiting time for inner core nucleation is warranted.

We have modified the text on lines 171-179 to better illustrate the meaning of these values and also added a section to the supplementary materials which provides a

more detailed description, including a case where the inner core is 300 Myrs old, showing that even for a more geophysically plausible scenario the outcome is similar due to the exponential relating in Eq 10. This description follows our recent review (Wilson et al., 2025, <https://doi.org/10.1038/s43017-024-00639-6>) which explores the topic in detail. Additionally, we have removed use of the work “moderate” because we agree with the reviewer that it is not helpful for the reader’s understanding of these values.

164-7. This is really pushing it. Even if the required supercooling at 10% C is just within the maximal bound on the core temperature change from nucleation to the present, it would require crystallization of essentially the entire inner core from a supercooled state, which would have occurred very rapidly, essentially yesterday, and with the “inner” inner core (if it represents early, rapid crystallization from a supercooled state) representing essentially the entire core. This is not a plausible scenario. The same goes, really, for the lower supercooling values discussed below, at higher C concentrations. In my opinion, the paper will be more effective if it’s up front about the plausible limits and doesn’t overstretch the argument for C.

The reviewer is correct, this notional agreement between the supercooling needed to nucleate from 10% C liquids and the maximum supercooling compatible with the structure of the core probably does not represent a situation that is realistic for inner core nucleation. We have explained this to be the case in the discussion and feel that this notional agreement is a demonstration that some core compositions can substantially reduce the supercooling requirement for homogeneous nucleation, showing that it could well be a viable mechanism for inner core nucleation.

Following the reviewer’s suggestion, the new text on lines 199-203 make a more direct statement on the significance of the maximum available supercooling and we have added a section to the supplementary information which give more details on the maximum supercooling which the core.

252. Certainly a novel constraint, with the potential to be strong. But there are still many uncertainties, including whether there is a plausible route via heterogeneous nucleation on a metallic substrate.

We thank the reviewer for their positive assessment of our finding. It is true that uncertainties do remain, including whether heterogeneous nucleation is a viable mechanism. Our comment that heterogeneous nucleation presents a reduction to the nucleation barrier, not a removal of it, makes the point that for heterogeneous nucleation to be viable, it must be underpinned by a homogeneous case which requires significantly less supercooling for nucleation than pure Fe and that despite previous efforts (Huguet et al., 2018, <https://doi.org/10.1016/j.epsl.2018.01.018>; Wilson et al., 2025, <https://doi.org/10.1038/s43017-024-00639-6>) no viable candidate for a heterogeneous nucleation site in the Earth’s liquid core has been identified thus far.

On lines 221-241 we note that heterogeneous nucleation does not remove the requirement for supercooling, and only a reduces it, and discuss the requirement of a suitable candidate heterogeneous nucleation site. On lines 294-299 we explain that the constraint on core composition proposed in our manuscript is still valid if heterogenous nucleation triggered inner core growth because of the requirement for a stable heterogeneous phase and a core composition compatible with nucleating onto this phase.

263-78. The Conclusion section consists mainly of points that are outside what is discussed in the main body of the paper, and not particularly dependent on the results presented.

We agree with the reviewer that in isolation the points discussed in the conclusion were

incongruent. We have opted to remove the conclusions section and incorporate these points into the final section of the discussion where they belong.

The modification can be found on lines 300-315.

Reviewer #3 (Remarks to the Author):

The manuscript builds on previous research investigating the supercooling behavior of core compositions relevant to Earth's core, specifically focusing on the binary Fe-C system. The authors employ molecular dynamics simulations to determine key parameters governing the transition from a fully liquid state to the onset of freezing. They then apply classical nucleation theory to interpret their findings and derive critical parameters, such as the required supercooling and the critical nucleus size.

The methodology, largely adopted from their previous work, has been well established in earlier studies. In this paper, they extend the approach to additional compositions within the Fe-C system, bringing their analysis closer to the supercooling conditions constrained by geophysical models. The central contribution of the manuscript, as I understand it, is to introduce a paradigm shift—positioning the study of supercooling behavior as an independent yet complementary constraint in planetary core research.

While the paper does not introduce significant methodological innovations (at least not entirely clear from manuscript), I recognize the authors' effort to highlight this new perspective. This work is timely, as the challenge of nucleating Earth's inner core was identified by Huguet et al. (2018) and has remained an open question since. The authors, through this and previous work, have steadily developed a novel and constructive approach to addressing this issue. In this context, the present manuscript represents a crucial step in establishing supercooling behavior as a key research focus in planetary science.

The paper is well-written, concise, and clear. In my opinion, it is suitable for publication. Below, we (ECR and me) provide some general remarks and minor comments that should be addressed before final acceptance.

We thank both reviewers for their detailed review of the manuscript. We are pleased that they consider the work to be of sufficient significance and quality for publication. We have addressed each of the points raised in the response below and feel that the changes made do improve the clarity of manuscript, especially for a broad audience.

General Comments:

- It is somewhat unfortunate that the simulations fail for $x > 0.1$, as these cases appear to be the most relevant in terms of geophysical constraints. It would be helpful to provide a more detailed explanation of the issues causing these failures. While there are hints throughout the text, a clearer and more concrete statement would be beneficial—does the problem stem from the limitations of the EAM potential? What are the implications of this failure?

We thank the reviewer for this comment. We observe phase separation in high x^C simulations, where the carbon rich phase has a higher x^C (>0.3) than the ab initio data used to develop and validate the potential. Therefore, further testing of this behaviour with comparison to equivalent ab initio data is needed for interpretation. This limitation does not impact this study because the atomic potential used in the MD calculations is shown to reproduce ab initio results for the x^C range of simulations which are retained in the study. We refer the reviewer to our response to reviewer 1's 4th comment for additional detail on why we decided not to produce an expanded ab initio dataset for validation of this behaviour. **We have added text to lines 158-162 of the results section to explain this behaviour and the reason that we omit results where $x^C > 0.1$ from our study.**

- The discussion of geophysical constraints could be expanded. What specific geophysical models are referenced? To what extent are they independent of the assumed concentrations? Given that supercooling behavior is being proposed as a new constraint on core composition, a more detailed explanation of these models and their role in this context would strengthen the manuscript.

The reviewer raises an important point. The absolute maximum supercooling of the inner core ($dT = 420$ K) from the model of our previous study (Wilson et al., 2023, <https://doi.org/10.1016/j.epsl.2023.118176>) is not self-consistent in that it does not require that the material properties of a particular core composition can explain both the extreme melting temperature and adiabatic gradient required for this value. However, whilst the absolute values of melting temperature and adiabatic temperature can be significantly altered by the composition of the core, the profile (or gradients) of T_m and T_a are relatively insensitive to composition. This means that the geophysically inferred maximum value of dT can be matched using many different compositions. The composition of the core is therefore not constrained by the location of the inner core boundary, but the nucleation process itself can provide a strong constraint, as our study shows.

The topic of evaluating the geophysical constraints on core supercooling does require lengthy explanation to be digestible. Since the fine details of this topic are not central to the message of this manuscript, we have included a section in the supplementary material which lays out these concepts in more detail and also references our recent review article (Wilson et al., 2025, <https://doi.org/10.1038/s43017-024-00639-6>) which includes an in-depth exploration.

We have added text at lines 195-199 to note that the composition of the core does not have a strong influence on the maximum supercooling of the core and have included a statement on how thermal evolution modelling can be used to evaluate the constraint imposed by geophysical observations on the supercooling of the core on lines 288-289 and a reference to the supplementary material on lines 200-201.

- The paper would benefit from a clearer distinction from the authors' previous work, particularly Wilson et al. (2023) [33]. It would be helpful to explicitly state any methodological differences and clarify how this study extends prior research (e.g., by applying the same methodology to mixtures with higher carbon concentrations). For instance, the direct reference to [33] in lines 99–101 could be expanded with a statement such as: "We extend previous studies by considering ..." to emphasize the novelty of the present work.

We thank the reviewer for pointing out this opportunity to clarify the advances of this study. **We have modified the text on lines 101-104 in line with the suggestions made.**

- Along the same lines: In what way do the contributions unique to this paper—such as the investigation of the F-C mixture at higher carbon concentrations — establish this admissibility criterion in a manner not previously considered?

We thank the reviewer for this suggestion and have added text to the beginning of the discussion section which explains how this study builds upon our previous study and brings us closer to identifying a mechanism which can explain inner core nucleation at supercooling which is compatible with geophysical observations of the deep Earth.

This change can be found on lines 185-188.

- Similarly, the discussion of results could be strengthened with a summarizing statement. For example: "Through the simulations of the Fe-C mixture at higher x conducted in this study, we have taken a first step toward assessing the undercooling admissibility criterion ..."

Following this comment and the those preceding the text on lines 183-185 is now a strengthened summarizing statement and a better introduction to the discussion of our findings.

Minor comments:

- I. 119: The definition of abbreviation EAM for “Embedded Atomic Model” is missing. It is only introduced in the later Sec. 5.

Thank you, this has now been **corrected on line 123.**

- I. 121: the variable x should probably be an $x^{\wedge}C$

Thank you, this has now been **in all instances.**

- Fig. 1: in the lower plot, the y axis label is missing a variable name (e.g. x or $x^{\wedge}C$)

Thank you, this has now been corrected **throughout.**

- I. 128: the nucleation rate $I(r)$ is used as a function. Since it was only mentioned once before repeating at least the name here would make it easier to read

Thank you, **text added at line 133.**

- I. 137: starting a sentence with a variable τ_w . Change to something like “The average waiting time τ_w is computed as $\tau_w = \tau_0 \exp(\dots)$ ”

Thank you, the suggest **change has been made on line 147.**

- I. 145: it is not easily traceable for me what the differences of the interatomic potential are wrt. to the existing work from Wilson et al. 2023. The methodology described in 5.1 seems to be exactly the same. For me, a highlight of differences would be useful to assess in which regard this work is new.

We now provide a more detailed account of the new potential validation and differences with our prior study. **These can be found in the new supplementary materials and lines 352-353.**

- I. 148: missing “break” between symbols δT and r_c , recommend to change to “at large δT critical radii r_c for all ...”

Correction made at line 158.

- II. 148-150: please elaborate a bit on the type of instability / why the method breaks down (see general comment above). It is fair the method has limitations but it would be helpful to understand them better

We have added text at line 158-162 to better describe the observed instability.

- I. 157: Waiting time is given in time/volume. It would be great to have one or two explanatory sentences as to how this relates to a time in the classical sense.

The new text on line 176-180 provides an additional means of intuitive interpretation for the waiting time quantity.

- I. 328: the abbreviation “NVE” has not been defined

Text has been added on line 366-367 to clarify this ensemble.

- I. 414: please fix the link to the zenodo repository to include a full copyable doi link, e.g. www.doi.org/10.5281/zenodo.13144422 . Also maybe rephrase the sentence such it does not end with the link to avoid merging the sentence-ending dot with the URL

This link has been updated and the format corrected.

Small comments (mostly writing):

- I.66: delete “result in ...” (duplication)

Corrected.

- I. 79: missing comma after “Here”

Corrected.

- I. 94: “Ref” shown before reference [34]

Corrected.

- I. 101: missing space before reference [33]

Corrected.

- I. 103: potential comma after “nucleation”

We believe the grammar is correct here.

- II. 129-130: repeated word “which”

Corrected.

- I. 147: missing comma before “although”

Corrected.

- Table 1: maybe put the unit of variables in its own column in the tabular to fix alignment

Whilst we agree that this may be a good choice, we are reluctant to make formatting changes prior to typesetting as the manuscript will go from a standard latex draft to a 2 column Nature Communications article. We will pay close attention to these details at the typesetting phase.

- I. 237: missing comma -> “which, when applied to a specific model of core formation, ...”

Corrected.

- II. 238-239: repeated use of the word “strongly”, suggest to replace once

Corrected.

- II. 346-347: double use of the word “consistent”

Corrected.

- I. 422: Section heading weirdly transitions into the following paragraph.

Corrected.

Reviewer #4 (Remarks to the Author):
